# Pleural Peels Tissue Culture plus Pleural Fluid Culture Help to Improve Culture Rate for Empyema

**DOI:** 10.3390/jcm11071882

**Published:** 2022-03-28

**Authors:** Ya-Fu Cheng, Ching-Yuan Cheng, Chang-Lun Huang, Wei-Heng Hung, Bing-Yen Wang

**Affiliations:** 1Division of Thoracic Surgery, Department of Surgery, Changhua Christian Hospital, Changhua 500, Taiwan; 181033@cch.org.tw (Y.-F.C.); 62980@cch.org.tw (C.-Y.C.); 144474@cch.org.tw (C.-L.H.); 161450@cch.org.tw (W.-H.H.); 2School of Medicine, Chung Shan Medical University, Taichung 201, Taiwan; 3School of Medicine, College of Medicine, Kaohsiung Medical University, Kaohsiung 807, Taiwan; 4Institute of Genomics and Bioinformatics, National Chung Hsing University, Taichung 402, Taiwan; 5Department of Post-Baccalaureate Medicine, College of Medicine, National Chung Hsing University, Taichung 402, Taiwan; 6Center for General Education, Ming Dao University, Changhua 523, Taiwan

**Keywords:** empyema, lung infection, pleura, thoracoscopy/VATS

## Abstract

Background: Empyema is known as a serious infection, and outcomes of empyema cases remain poor. Pleural fluid culture and blood culture have been reported to give unsatisfactory results. We introduce a novel pleural peels tissue culture during surgery and aim to improve the culture results of empyema. Methods: This was a retrospective study and was obtained from our institute. Patients with stage II or III empyema undergoing video-assisted thoracic surgery decortication from January 2019 to June 2021 were included in the study. Results: There were 239 patients that received a pleural peels tissue culture, a pleural fluid culture, and a blood culture concurrently during the perioperative period. Of these, 153 patients had at least one positive culture and 86 patients showed triple negative culture results. The positive culture rates were 46.9% for pleural peels tissue cultures, 46.0% for pleural fluid cultures, and 10% for blood cultures. The combination of pleural peels tissue culture and pleural fluid culture increased the positive rate to 62.7%. Streptococcus species and Staphylococcus species were the most common pathogens. Conclusion: The combination of pleural peels tissue culture and pleural fluid culture is an effective method to improve the positive culture rate in empyema.

## 1. Introduction

Empyema is known as a serious infection with poor outcomes. Its incidence has tripled in the past decade [1]. Previous studies reported that the 30-day mortality rate was around 7% to 9% for patients with thoracic empyema undergoing decortication of the pleura [2,3]. The Thoracic Surgeons’ General Thoracic Surgery Database also showed that complications occurred around 39% of the time and the readmission rate was 8.7% [4].

Empyema has been classified into three stages: (1) exudative phase, (2) fibrinopurulent phase, and (3) organizing phase. In stage I there is a clear and sterile pleural effusion; stage II involves pleural peels formation with infected and purulent pleural effusion; stage III is the formation of thick granulation tissue and noticeable lung encasement. During stage I, antibiotics treatment is adequate and chest tube drainage is not required [5]. If the parapneumonic effusion progresses to stage II, then the tissue type plasminogen activator (tPA) level decreases and there is fibrin deposition and septation formation. Surgical intervention with chest tube insertion is recommended in this stage. Decortication of pleural peels is suggested to support the re-expansion of the lung encasement. Previous studies revealed that decortication via video-assisted thoracic surgery (VATS) has fewer postoperative complications compared to open thoracotomy. It also offers the same rate of success [3,6].

To improve the outcomes of empyema, early surgical intervention and proper antibiotics usage are both essential [7]. A higher positive culture result leads to proper antibiotics usage. Previous data showed less than 12% of blood cultures have been reported as positive [8]. The pleural fluid cultures have positive rates ranging from 19% to 49% [9,10]. There were no studies reporting the positive rate of pleural peels tissue cultures during surgery or the results of the multiple culture method.

## 2. Materials and Methods

### 2.1. Patient Population and Selection

Our study was a retrospective analysis in our institute (Changhua Christian Hospital, Changhua, Taiwan) and was approved by our institutional review board (IRB-210808). Informed consent from all participants was waived. All patients over 18 years of age with thoracic empyema undergoing thoracoscopic decortication of the pleura from January 2019 to June 2021 were included in the study. Patients with stage I empyema or missing lab data and patients lost to follow-up were excluded. Those patients for whom pleural peels tissue culture, pleural fluid culture, and blood culture were not collected together perioperatively were also excluded. No matter what pleural fluid culture was positive or negative pre-operatively, we obtained both pleural peels tissue culture and pleural fluid culture intraoperatively. We do aerobic culture, anaerobic culture, mycobacteria and fungal cultures routinely for both pleural fluid culture and pleural peels tissue culture. Figure 1 shows the pleural peels tissue culture in a patient with stage II empyema.

We analyzed the age, gender, empyema location, empyema stage, cause of empyema, antibiotics usage, lab data, pleural fluid data, length of hospital stay, pathogen and mortality. Mortality was defined as death during hospitalization or within thirty postoperative days. The outcome measures for our study were culture positive rate and the type of pathogen. Every observation was defined according to the American Association for Thoracic Surgery (AATS) consensus guidelines released in 2017. A pleural fluid pH value less than 7.2, pleural fluid glucose less than 40 mg/dL, and pleural fluid lactate dehydrogenase (LDH) greater than 1000 IU/L are indicators of pleural space infection and predictors of a complicated clinical course.

### 2.2. Statistical Analyses

The following clinic-pathologic factors were included into analyses: age, gender, empyema location, empyema stage, cause of empyema, antibiotics usage, lab data, pleural fluid data, and length of hospital stay. We used the mean and standard deviation (SD) to evaluate the continuous variables in this study. As for the categorical variables, we used the chi-squared test or Fisher’s exact test to compare the numbers and proportions. 

We also used adjusted multinomial logistic regression analysis to determine the odds ratio of a positive culture rate. The variables that were significantly different between the two groups, as well as the variables that were associated with the positive culture rate, were selected into the maximum model. They were controlled and adjusted together to eliminate the effects of confounders. All calculations were performed using IBM SPSS Statistics for Windows, Version 22.0 (IBM Corp., Armonk, NY, USA). Statistical analysis with a *p*-value less than 0.05 was considered statistically significant.

## 3. Results

Two hundred and thirty-nine patients with stage II to stage III thoracic empyema underwent thoracoscopic decortication in this study. From these patients, a pleural peels tissue culture, a pleural fluid culture, and a blood culture were all collected during the perioperative period. For 153 patients, at least one culture revealed a positive result, and 86 patients showed triple negative culture results in this study. 

The clinic-pathological characteristics of all study patients are shown in Table 1. The mean age was about 63 years old. Most of the patients were males (77.8%). The location of empyema was predominantly on the right side (64.9%). Stage II empyema (76.6%) was more common than stage III empyema (23.4%). Over ninety percent of the empyema cases were caused by pneumonia. Other cases were followed by abdominal origin, esophageal origin, iatrogenic, and trauma. The mean preoperative antibiotics usage was about 5 days, and the 30-day mortality rate was 7.1%. Of the 239 total patients, 43.9% showed a single pathogen and 20.1% revealed multiple pathogens. 

The distribution of the pathogens is shown in Figure 2. Our study revealed that most common species of pathogens in empyema was Streptococcus (26%). It was followed by Staphylococcus (13%), Klebsiella (6%), Parvimonas (5%), Paenibacillus (5%), and Pseudomonas (5%). The most common fungal infection species was Candida (2%). Mycobacteria was found in about 4% of the empyema cultures. 

Figure 3 shows the positive culture rate in each culture. The pleural peels tissue cultures during operation revealed a 46.9% positive culture rate. The perioperative pleural fluid positive culture rate was similar (46.0%). The blood cultures merely had a 10% positive culture rate. Concurrently using a pleural peels tissue culture and a pleural fluid culture significantly elevated the positive culture rate to 62.7%. The use of all three culture methods at the same time resulted in a positive culture rate of 64.0%. 

We divided the patients into a positive culture group and a negative culture group. We tried to figure out the influencing factor of the culture results (Table 2). The age distribution was similar in both groups (63.56 years in the positive culture group vs. 63.06 years in the negative culture group, *p* = 0.820). The proportion of males was also similar in both groups (79.7% in the positive culture group vs. 74.4% in the negative culture group, *p* = 0.342). Both groups were right side (66.7% in the positive culture group vs. 61.6% in the negative culture group, *p* = 0.434) and stage II (77.8% in the positive culture group vs. 74.4% in the negative culture group, *p* = 0.556) predominant. 

Regarding preoperative antibiotics usage, it is clear that a shorter time of antibiotics usage before operation correlated with a higher positive culture rate. The mean preoperative antibiotics usage was 4.02 days in the positive culture group versus 6.87 days in the negative culture group (***p* = 0.002**). 

It is interesting to find that a positive culture was much more related to absolute neutrophil count (ANC) instead of white blood cell (WBC) count. Although the WBC count was higher in the positive culture group (14,331 /μL vs. 12,941 /μL, *p* = 0.100), it was not statistically significantly different. The ANC was a much better gauge to predict culture results (12,351 /μL in positive culture group vs. 10,432 /μL in positive culture group, ***p* = 0.016**). Pleural fluid data was also strongly related to the prediction of culture results. A pleural pH count less than 7.2 (62.7% vs. 46.5%, ***p* = 0.015**), pleural glucose less than 40 mg/dL (45.8% vs. 24.4%, ***p* = 0.001**), and pleural LDH more than 1000 (66.0% vs. 45.3%, ***p* = 0.002**) were characteristics of patients who were more likely to obtain positive cultures.

We used adjusted multinomial logistic regression analyses to find the odds ratio (OR) of each factor for prediction of the culture result (Table 3). We divided the culture result into single culture positive and multiple culture positive (stronger prediction) groups. We suggested that preoperative antibiotics usage, pleural fluid glucose, and pleural fluid LDH were the most important factors. The positive culture rate decreased 10% for each day that a preoperative antibiotic was used. When pleural fluid glucose was less than 40 mg/dL, there was a 2.5 times greater positive culture rate. 

## 4. Discussion

Our study is a retrospective study, and we suggest that the pleural peels tissue culture (46.9%) obtains a similar result as the pleural fluid culture (46.0%). The combination of pleural peels tissue culture and pleural fluid culture supported a higher positive culture rate (62.7%). Blood culture is less important in empyema. Understanding these outcomes is essential for surgeons to select proper culture methods during decortication for empyema. The treatment of empyema has an emphasis on achieving adequate evacuation of the effusion and lung re-expansion [11]. Proper antibiotics usage for pathogens according to the culture result is also vital.

Several factors were associated with the culture results. The preoperative antibiotics usage duration had the largest influence on the culture results. Our data revealed that the positive culture rate decreased by 10% for every extra day of preoperative antibiotics use. Early intervention with proper culture methods results in a higher positive culture rate. A previous study also suggested that obtaining blood cultures during antibiotic therapy is associated with a significant loss of pathogen detection. It reduced the positive culture rate by 20% during antibiotic therapy [12]. A prompt culture before antibiotics introduction is imperative for patients with empyema.

All patients in this study underwent VATS decortication rather than thoracotomy. There were several studies that compared VATS decortication to open surgery. Despite the resolution of disease being equivalent in both stage II and III empyema, the perioperative outcomes are different [13]. VATS is associated with shorter operative time, less pain, reduced hospital stays, and lower perioperative morbidity and mortality [14,15,16]. With the progress of VATS techniques, the conversion rates are lower in recent studies. However, VATS may not always be the ‘‘best’’ approach [4]. Late stage chronic empyema with diffuse fibrotic change is suitable for open surgery. Early surgical intervention to avoid late-stage empyema is crucial to provide better outcomes.

The pathogen spectrum varies by study and geographical area. Most studies indicated that Streptococcus species and anaerobes were in the majority of community-acquired pneumonia cases and Staphylococcus species were in the majority of hospital-acquired pneumonia cases [17,18]. However, recent research indicated that the anaerobes and Staphylococcus species have gradually replaced Streptococcus species as the major pathogens in surgically treated empyema [19]. Our study indicated that Streptococcus species and Staphylococcus species accounted for about 40% of the pathogens in empyema. There are still variable pathogens needing to be covered by antibiotics, and over 20% of patients are infected by multiple pathogens. Polymicrobial infection is common with Gram-negative organisms and anaerobes. It is more frequent in elderly patients and those with a comorbid disease [20]. Initial broad-spectrum antibiotics usage and specific antibiotics based on the culture result are warranted for the treatment of empyema [21]. Fungal empyema is rare and often accounts for less than 1% of pleural infections. The majority of fungal empyema cases are of the Candida species, and the mortality rate can be up to 73% [22,23]. Our study comes to the same conclusions, and we suggest that fungal empyema should be considered if the infection progresses after prolonged antibiotics treatment.

To try to elevate the culture positive rate, we used 16S rRNA PCR for thoracic empyema. This technique has been applied since 2019, but the outcomes seemed to be insufficient in characterizing the microbial spectrum of empyema [24]. However, Temi et al. applied 16 S rRNA PCR on 90 patients with culture-negative pleural fluid specimens. Thirty-one percent of the culture-negative pleural fluid specimens yielded a positive PCR result [25]. The effect of 16S rRNA PCR in confirming pleural infection is still controversial and should be further explored.

This study is the first study to report the positive culture rate of pleural peels tissue cultures in empyema. We demonstrate that the pleural peels tissue culture is useful and improves the positive culture rate significantly when it is combined with the pleural fluid culture (46% to 62%). We suggest to initially introduce broad-spectrum antibiotics to cover Streptococcus species, Staphylococcus species, and certain Gram-negative organisms while collecting a prompt culture. There are some limitations of our study. First, this is a retrospective study that may have selection bias, and this could affect the data analysis. Second, the single center study did not have sufficient evidence to be applicable to every institute. The pathogens and culture techniques may vary slightly among different places. A large-scale multi-center study is needed to confirm our results. Though there are several limitations, the introduction of the pleural peels tissue culture to patients with empyema is an innovation and an effective method.

## 5. Conclusions

The combination of pleural peels tissue culture and pleural fluid culture seems to be an effective method to improve the positive culture rate in empyema. A prompt culture before antibiotics introduction and early surgical intervention are the keys to improve the outcome of empyema.

## Figures and Tables

**Figure 1 jcm-11-01882-f001:**
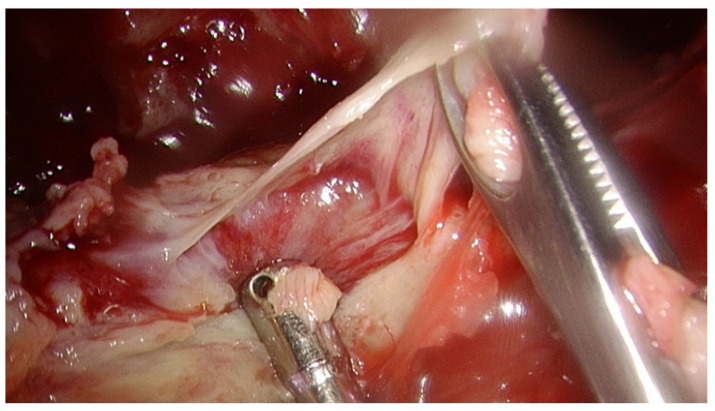
The pleural peels tissue culture in a patient with stage II empyema.

**Figure 2 jcm-11-01882-f002:**
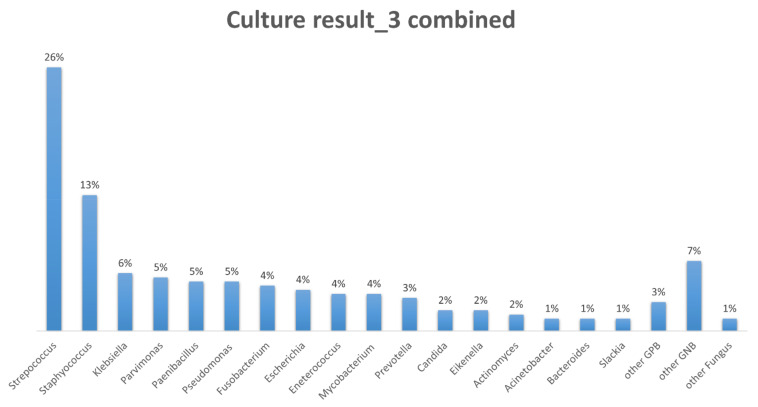
The distribution of the pathogens. Streptococcus (26%) and Staphylococcus (13%) are the most common species of pathogens in empyema.

**Figure 3 jcm-11-01882-f003:**
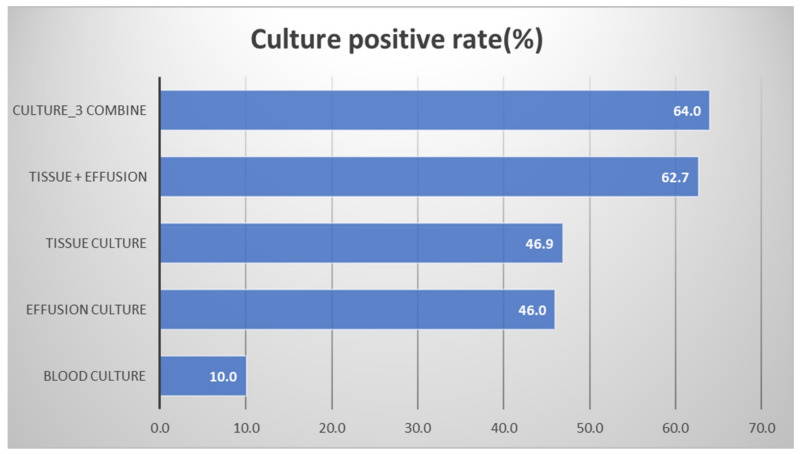
The positive culture rate in each culture.

**Table 1 jcm-11-01882-t001:** Clinical demographic data of patients.

Factors (Mean ± SD)	Total Cohort
Number of patients	239
Age (years)	63.38 ± 16.34
Gender	
Male	186 (77.8%)
Female	53 (22.2%)
Location	
Right	155 (64.9%)
Left	84 (35.1%)
Stage	
II	183 (76.6%)
III	56 (23.4%)
Cause	
Pneumonia	218 (91.2%)
From abdomen	12 (5.0%)
From esophagus	5 (2.1%)
Iatrogenic	3 (1.3%)
Trauma	1 (0.4%)
Pre-OP ABX usage (days)	5.04 ± 5.79
Length of hospital stay	28.72 ± 36.78
Mortality in 30 days	17(7.1%)
Pathogen	
Single	105 (43.9%)
Multiple	48 (20.1%)

OP: operative; ABX: antibiotics.

**Table 2 jcm-11-01882-t002:** Factors to predict the combined results of three types of cultures (pleural peels tissue, pleural fluid, and blood).

Factors (Mean ± SD)	At Least One Positive Culture (n = 153)	Three Negative Cultures (n = 86)	*p*-Value
Age (years)	63.56 ± 16.64	63.06 ± 15.88	0.820
Gender, male	122 (79.7%)	64 (74.4%)	0.342
Location, right	102 (66.7%)	53 (61.6%)	0.434
Stage, II	119 (77.8%)	64 (74.4%)	0.556
Pre-OP ABX usage (days)	4.02 ± 3.61	6.87 ± 8.08	**0.002**
Lab data			
WBC (/μL)	14,331.76 ± 6366.75	12,941.16 ± 6016.63	0.100
ANC (/μL)	12,351.13 ± 6017.91	10,432.57 ± 5573.57	**0.016**
Pleural data			
Pleural pH ≦ 7.2	96 (62.7%)	40 (46.5%)	**0.015**
Pleural glucose ≦ 40 mg/dL	70 (45.8%)	21 (24.4%)	**0.001**
Pleural LDH ≧ 1000 IU/L	101 (66.0%)	39 (45.3%)	**0.002**

SD: standard deviation; WBC: white blood cells; ANC: absolute neutrophil count; LDH: lactate dehydrogenase.

**Table 3 jcm-11-01882-t003:** Factors to predict the combined results of three types of cultures (pleural peels tissue, pleural fluid, and blood).

	Single Culture Positive		Multiple Culture Positive	
	OR (95% CI)	*p*	OR (95% CI)	*p*
Age	1.29 (0.80~2.08)	0.288	1.16 (0.73~1.85)	0.534
Gender (female)	0.53 (0.22~1.29)	0.162	0.81 (0.37~1.80)	0.604
ANC	1.00 (0.99~1.01)	0.472	1.00 (1.00~1.00)	0.968
Preoperative antibiotic duration	0.88 (0.80~0.95)	**0.003**	0.90 (0.83~0.97)	**0.006**
Location (left)	0.53 (0.25~1.13)	0.099	0.64 (0.31~1.31)	0.220
Stage (II)	2.71 (1.08~6.78)	**0.033**	1.48 (0.67~3.26)	0.333
Pleural pH ≦ 7.2	1.34 (0.64~2.80)	0.435	1.40 (0.69~2.84)	0.357
Pleural glucose ≦ 40	1.82 (0.79~4.22)	0.162	2.50 (1.15~5.42)	**0.020**
Pleural LDH ≧ 1000	1.41 (0.66~2.99)	0.373	2.12 (1.02~4.41)	**0.045**
Length of hospital stay	1.02 (1.00~1.04)	**0.014**	1.02 (1.00~1.04)	0.054

OR: odds ratio; CI: confidence interval; ANC: absolute neutrophil count; LDH: lactate dehydrogenase.

## Data Availability

Not applicable.

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
