# Peer review of "Pleural Peels Tissue Culture plus Pleural Fluid Culture Help to Improve Culture Rate for Empyema"

_jcm, 2022, doi:10.3390/jcm11071882_

Round 1

Reviewer 1 Report

Authors performed a thoracoscopic decortication of the pleura in thoracic empyema patients for whom pleural peels tissue culture, pleural fluid culture, and blood culture were collected together perioperatively. a combination of pleural peels tissue culture and pleural fluid culture to improve the positive culture rate in empyema. And they concluded that the combination of pleural peels tissue culture and pleural fluid culture is an effective method to improve the positive culture rate in empyema.

I would like to suggest some points to improve this paper.

Minor comments:

  1. (line 38) 

5 In stage I there is a clear and sterile pleural effusion;

 ---> In stage I there is a clear and sterile pleural effusion;

  1. (line 74) 

pleural fluid less than 40 mg/dL

 --- what is the meaning?

  1. (line 100-101) 

Other cases were of abdominal or esophageal origin, and one case was iatrogenic.

 --- Three cases were iatrogenic in Table 1.

  1. (line 132) 

WBC: white blood cells; ANC: absolute neutrophil count; LDH: lactate dehydrogenase.

 ---> SD: standard deviation; WBC: white blood cells; ANC: absolute neutrophil count; LDH: lactate dehydrogenase.

Author Response

Reviewer 1 comments

Comment 1:

    (line 38)

5 In stage I there is a clear and sterile pleural effusion;

 ---> In stage I there is a clear and sterile pleural effusion;

Answer 1:

It’s our mistake to plus additional “5” at line 38. We have deleted it. Thanks for the reminder!

Changes 1:

   Line 38

Comment 2:

(line 74)

pleural fluid less than 40 mg/dL

--- what is the meaning?

Answer 2:

It’s our mistake to mistype “pleural fluid glucose less than 40 mg/dL” to “pleural fluid glucose less than 40 mg/dL”. We have corrected it. Thanks for the reminder!

Changes 2:

   Line 74

Comment 3:

(line 100-101)

Other cases were of abdominal or esophageal origin, and one case was iatrogenic.

--- Three cases were iatrogenic in Table 1.

Answer 3:

  The Table 1. is correct! It’s our mistake to mistype the data. We corrected “Other cases were of abdominal or esophageal origin, and one case was iatrogenic.” to “Other cases were followed by abdominal origin, esophageal origin, iatrogenic and trauma.” Thanks for the reminder!

Changes 3:

  Line 100-101

Comment 4:

(line 132)

WBC: white blood cells; ANC: absolute neutrophil count; LDH: lactate dehydrogenase.

 ---> SD: standard deviation; WBC: white blood cells; ANC: absolute neutrophil count; LDH: lactate dehydrogenase.

Answer 4:

  We have added “SD: standard deviation” to Table 2. Thanks for the reminder!

Changes 4:

  Line 132

Reviewer 2 Report

This is a retrospective study of patients undergoing decortication due to empyema. The authors show that among the patients in which blood, pleural fluid and pleural peel cultures were sent, the addition yield of the peel cultures was significant for identifiyng a pathogen. 

There are some limitations of the study:

  1. Only patients with data regarding these three cultures were included in the analysis and not all patients undergoing surgery. This may bias the results of the yield of each culture since perhaps the peel culture was only sent in patients where the fluid culture was negative pre-operatively (assuming the pre-operative fluid culture was indeed analyzed). It would be more complete to see the entire database analysis.
  2. The analysis of positive cultures was done even when the cultures revieled micro-organism that are likely non-pathogenic (i.e candida). 
  3. We do not know the yield of fluid culture positivity in Empyma cases not going for surgery in this institute. If it is high (because usually taken closer to initiating antibiotics), perhaps the peel culture is not neccessary.   

In the methods section please indicate if routine cultures were sent for mycobacteria and fungal cultures via all methods, or if this was not routine practice, as this can affect the results.  

In addition, the paper needs some English proofing. 

Round 2

Reviewer 2 Report

Thank you for addressing the comments of my review.

Author Response

Q: Thank you for addressing the comments of my review.

A: Thank you for the detailed review of our manuscript. There are many useful suggestions and make it better.